Beyond gene ontology (GO): using biocuration approach to improve the gene nomenclature and functional annotation of rice S-domain kinase subfamily

http://orcid.org/0000-0001-7819-4552 Naithani Sushma naithans@science.oregonstate.edu
http://orcid.org/0000-0001-8644-2632 Dikeman Daemon
Garg Priyanka
Al-Bader Noor
http://orcid.org/0000-0002-1005-8383 Jaiswal Pankaj
Botany and Plant Pathology, Oregon State University , Corvallis, OR , USA
Maloof Julin
Electronic publication date: 2021 Mar 15
Publication date: 2021
Volume: 9
Electronic Location ID: e11052
Received 2020 Jul 23; Accepted 2021 Feb 11
Copyright: © 2021 Naithani et al.
Copyright year: 2021
Copyright holder: Naithani et al.
License: This is an open access article distributed under the terms of the Creative Commons Attribution License, which permits unrestricted use, distribution, reproduction and adaptation in any medium and for any purpose provided that it is properly attributed. For attribution, the original author(s), title, publication source (PeerJ) and either DOI or URL of the article must be cited.
License URL: https://creativecommons.org/licenses/by/4.0/

Keywords: Receptor-like kinase, SD-RLK, Abiotic stress response, Biotic stress, Rice, Oryza sativa, Plant development, Data-mining, Secondary data analysis, Gene nomenclature for the rice SDRLK family

Funding: National Science Foundation NSF IOS-1127112 This work was supported by the Oregon State University funds to Sushma Naithani and National Science Foundation award to Gramene project (NSF IOS-1127112). The funders had no role in study design, data collection and analysis, decision to publish, or preparation of the manuscript.

==============================
The S-domain subfamily of receptor-like kinases (SDRLKs) in plants is poorly characterized. Most members of this subfamily are currently assigned gene function based on the S-locus Receptor Kinase from Brassica that acts as the female determinant of self-incompatibility (SI). However, Brassica like SI mechanisms does not exist in most plants. Thus, automated Gene Ontology (GO) pipelines are not sufficient for functional annotation of SDRLK subfamily members and lead to erroneous association with the GO biological process of SI. Here, we show that manual bio-curation can help to correct and improve the gene annotations and association with relevant biological processes. Using publicly available genomic and transcriptome datasets, we conducted a detailed analysis of the expansion of the rice (Oryza sativa) SDRLK subfamily, the structure of individual genes and proteins, and their expression.The 144-member SDRLK family in rice consists of 82 receptor-like kinases (RLKs) (67 full-length, 15 truncated),12 receptor-like proteins, 14 SD kinases, 26 kinase-like and 10 GnK2 domain-containing kinases and RLKs. Except for nine genes, all other SDRLK family members are transcribed in rice, but they vary in their tissue-specific and stress-response expression profiles. Furthermore, 98 genes show differential expression under biotic stress and 98 genes show differential expression under abiotic stress conditions, but share 81 genes in common.Our analysis led to the identification of candidate genes likely to play important roles in plant development, pathogen resistance, and abiotic stress tolerance. We propose a nomenclature for 144 SDRLK gene family members based on gene/protein conserved structural features, gene expression profiles, and literature review. Our biocuration approach, rooted in the principles of findability, accessibility, interoperability and reusability, sets forth an example of how manual annotation of large-gene families can fill in the knowledge gap that exists due to the implementation of automated GO projections, thereby helping to improve the quality and contents of public databases.

Introduction

Receptor-like kinases (RLKs) are major players in perceiving and transducing extracellular signals into appropriate cellular responses and have been associated with nearly every aspect of plant growth and development (Becraft, Stinard & McCarty, 1996; Berckmans et al., 2020; Cartwright, Humphries & Smith, 2009; Clark, Williams & Meyerowitz, 1997), organ differentiation (Hord et al., 2006; Jinn, Stone & Walker, 2000; Pu et al., 2017; Shpak, Lakeman & Torii, 2003), plant reproduction (Ahmadi et al., 2016; Escobar-Restrepo et al., 2007; Kachroo et al., 2001; Stein et al., 1991; Yu et al., 2016) and plant’s response to biotic (Chen et al., 2006; Fan et al., 2018; Gomez-Gomez & Boller, 2000; Pruitt et al., 2015), and abiotic stresses (Chen et al., 2013; Grison et al., 2019; Ouyang et al., 2010; Pan et al., 2020). The RLKs are encoded by one of the largest gene family, the RLK gene family (e.g., ~600 members in Arabidopsis thaliana and ~1,131 in Oryza sativa) (Lehti-Shiu & Shiu, 2012; Shiu et al., 2004).

Typically, an RLK protein consists of an extracellular N-terminal domain followed by a transmembrane domain, and a conserved C-terminal kinase domain. The extracellular domain of RLKs are involved in sensing the extracellular signals via interacting with ligands (i.e., small molecules such as hormones, glycoproteins, or short peptides), and the cytoplasmic Serine/Threonine (Ser/Thr) kinase domain transfers the signal by phosphorylating downstream targets to activate/inactivate the relevant physiological pathways (Wolf, 2017). RLKs display great variety in their extracellular domains, consistent with their role in recognizing the diverse ligands, whereas their kinase domains show high conservation in sequence and structure. Thus, based on the structure of their extracellular domains and the phylogenetic relationships between kinase domains, RLKs are grouped into subfamilies, such as leucine-rich repeats, S-domains (SD), Wall-associated kinases, etc. (Shiu & Bleecker, 2001a, 2001b).

The S-domain subfamily of receptor-like kinases (SDRLKs), the second-largest subfamily of RLKs, share similarity with the extracellular domain (or S-domain) of the S-locus receptor kinase (SRK) protein (Shiu & Bleecker, 2001a, 2003). SRK was first identified in Brassica stigma epidermis (Stein et al., 1991), where it acts as the female determinant of self-incompatibility (SI) during pollen-pistil interaction in an allele-specific manner (Kachroo et al., 2001). The ligand for SRK is a small peptide, known as S-locus cysteine-rich protein (SCR/SP11), located on the pollen surface (Naithani, Dharmawardhana & Nasrallah, 2013). A polymorphic S-locus region in the Brassica genome encodes both the SRK and SCR (Kachroo et al., 2001; Naithani, Dharmawardhana & Nasrallah, 2013; Takayama & Isogai, 2003). Upon pollination, SRK interacts with SCR, and a productive interaction leads to activation of the downstream signaling pathway that inhibits pollen germination. Thus, in the Brassicaceae family, selfing is prevented by SI and regulated by the S-locus haplotypes. Besides ligand binding (Boggs et al., 2009), the S-domain of SRK is also involved in receptor dimerization (Naithani et al., 2007) and pistil development (Tantikanjana et al., 2009).

Typically, after genome sequencing, based on similarity with SRK, the automated Gene Ontology (GO) pipelines assign electronic annotation to most SDRLK subfamily members as “SRK” or “S-domain receptor-like kinase” or “Similar to SRK protein”, and associate with the biological process of “SI” or “recognition of pollen” irrespective of their relevance in the given plant species. The partial genes of this family are assigned unknown functions. However, the sporophytic SI mechanism does not exist in rice and many other plant species. Recent studies of a few additional genes suggest that this subfamily members act as key regulators in diverse biological processes. For example, in rice Stress-Induced Kinase 2 (OsSIK2) (Chen et al., 2013), OsESG1 (Pan et al., 2020), Large Spike Kinase 1 (OsLSK1) (Zou et al., 2015), Plant Stature Related Kinase 2 (PSRK2) (Li et al., 2018) genes have been shown to play crucial roles in plant development. Arabidopsis CALMODULIN-BINDING RECEPTOR-LIKE PROTEIN KINASE1 (CBRLK1) (Kim et al., 2009), rice PID2 (Chen et al., 2006; Wang et al., 2015), and SPL11 Death Suppressor 2 (SDS2) contribute to pathogen resistance and immunity (Fan et al., 2018). Also, rice genes OsSIK2 (Chen et al., 2013), OsESG1 (Pan et al., 2020) have been shown to confer abiotic stress tolerance. Therefore, annotations of SDRLK subfamily genes in the genomic databases are erroneous and is liable to mislead researchers conducting high-throughput genomic analyses. Often, the SDRLK family genes are ignored as candidate genes for further experimental testing.

Accordingly, we propose improved gene nomenclature and descriptions for 144 members of rice (O. sativa) SDRLK subfamily genes using biocuration approach that involves collection of information about their structural features, transcription profiles, and subcellular localization of genes products from literature and based-on the analysis of publicly available genomics datasets. The improved gene annotation of rice SDRLK subfamily will help rice researchers conducting large-scale genomic analyses in identifying candidate genes involved in plant developmental processes and stress-response. We are the secondary users of the publicly available and/or previously published rice datasets. However, we are re-using these datasets to ask new questions to define structure and expression profile of rice SDRLK subfamily genes. Mostly, the primary generators of big genomic data do not fully utilize their data beyond their limited focus on a pathway or biological processes. Therefore, genomic datasets available under findability, accessibility, interoperability and reusability (FAIR) agreements are a gold mine for other researchers. A growing amount of data mining and secondary analyses of public datasets is recognized by the Pacific Symposium on Biocomputing, which celebrates the impactful meta-analyses through their annual Research Parasite Awards (https://researchparasite.com). Considering the prevalence of large gene families in plant genomes, our approach of re-use and re-analysis of publicly available genomic datasets and manual biocuration could be implemented for improving the annotation of other gene families.

Methods

Updates on rice SDRLK subfamily

Previously, 170 SDRLK subfamily members had been identified in rice (Oryza sativa ssp. japonica cv. Nipponbare) genome (Lehti-Shiu et al., 2009; Shiu et al., 2004) based on the gene models provided by the Michigan State University Rice Genome Annotation Project (MSU) project (http://rice.plantbiology.msu.edu) (27). Over the past 10 years, the reference rice gene models have been extensively revised, and the rice research community has adopted improved gene models and gene nomenclature from the Rice Annotation Project (RAP) (https://rapdb.dna.affrc.go.jp) (28). Thus, we revisited models for all 170 genes of the SDRLK family in the MSU dataset, and after removing obsolete, and short genes (encoding peptides of <150 amino acid and lacking any conserved protein domain), and including split and fused gene models, we proceeded with 144 genes for further analysis (see Table S1).

Gene structure and protein domain analysis

We used the rice genome browser provided by the Gramene (Tello-Ruiz et al., 2018) to investigate gene and transcript models of 144 SDRLK family members. For genes associated with alternatively spliced transcript isoforms, only the longest canonical transcript was considered. The information about protein domains and GO annotations was retrieved from the protein knowledgebase UniProt (https://www.uniprot.org). The information about the subcellular location of proteins was collected from the Compendium of Crop Proteins with Annotated Locations (http://crop-pal.org) and/or UniProt. If available, the gene/protein structure and/or localization information was further updated based on published studies (e.g., Os06g0494100, Os01g0890100) (Chen et al., 2006; Li et al., 2018). Table S1 provides information on the 144 rice SDRLK family (i.e., gene structures, protein domains, subcellular locations and associated GO biological processes).

Tandem gene duplication analysis

We extracted sequences of 144 rice SDRLK family proteins from the Gramene (http://gramene.org). We then built a sequence similarity tree to identify subgroups of genes that are most similar using Clustal Omega (https://www.ebi.ac.uk/Tools/msa/clustalo) (Sievers et al., 2011) software (set to default parameters). The tree file saved from Clustal Omega was exported to the Interactive Tree of Life (iToL; https://itol.embl.de) (Letunic & Bork, 2016) website to visualize and add data layers. We further investigated if genes that show high sequence similarity are in the same genomic neighborhood. We uploaded Gene IDs on the O. sativa genome browser available at Gramene and then manually inspected the genomic location and flanking regions of individual genes. We defined tandem duplicates based on sequence similarity and genomic location (<5 genes in between).

Gene expression analysis and visualization

The publicly available baseline and differential gene expression data of the rice SDRLK family genes used in this study and relevant information (for example, tissue sample, growth and developmental stage of plants, stress treatment and source) is provided in Table S2. All datasets used in this study contain high quality sequence data with minimum three replicates.

The baseline tissue-specific gene expression data for rice SDRLK genes was extracted from E-GEOD-50777 (Anderson et al., 2013), E-MTAB-4123 (Zhang et al., 2014), E-MTAB-2037 (Sakai et al., 2011) and E-MTAB-2039 (Davidson et al., 2012) datasets available from EMBL-EBI Expression Atlas (www.ebi.ac.uk/gxa/plant/experiments). All four datasets are from O. sativa ssp. japonica rice plants grown under normal growth conditions, and together represent expression data for SDRLK family genes from various vegetative and reproductive tissues. The Integrated RNA-Seq Analysis Pipeline (iRAP) developed by EBI Gene Expression Atlas was used for the data analysis including FASTQ QC, alignment, mapping QC and gene expression quantification. Each experiment is manually curated to a high standard and mapped to the most recent assembly of the rice reference genome. The tools employed for genome alignment, quantification and differential expression include Tophat2/Star, HTSeq-count (Papatheodorou et al., 2020). The normalized gene expression values are for all samples in each dataset are in Transcripts Per Million (TPM) (see Table S3). The data from all four files were plotted side-by-side while individual datasets were maintained as an independent block (samples were not clustered based on tissue types). Considering that each tissue or cell-type differs significantly from each other in their mRNA population, we did not make direct quantitative comparisons between the samples and across the datasets. But, made a comparison of expression profile among the SDRLK genes within each tissue sample to summarize baseline tissue-specific expression.

We found five differential gene expression (log2-fold change 1.0 and adjusted p ≤ 0.05) datasets for biotic stress treatments. E-GEOD-61833 (Girija et al., 2017); E-GEOD-36272; E-MTAB-5025 (Zhao et al., 2016); E-MTAB-4406 (Magbanua et al., 2014) datasets are available at the EMBL-EBI Expression Atlas and have been analyzed by the iRAP pipeline similar to baseline tissue-specific data described above, and then subjected to DESeq2 for identifying differentially expressed genes (DEG). The E-MTAB-6402 (ERS2106879–ERS2106902) was previously generated in our laboratory using Illumina HiSeq 3000 (Illumina Inc., San Diego, CA, USA). The processing of raw sequence data, read alignment to the rice japonica reference cv. Nipponbare IRSPG 1.0 genome using the HTSeq2 and the DESEq2 (similar to the iRAP) was used for identification of DEG in response to R. solani infection in MCR and CCDR varieties (cutoff adjusted P-value of ≤ 0.05). Also, we extracted microarray expression data for 14 SDRLK family genes from a published study (Narsai et al., 2013) describing differential expression of genes in response to Xanthomonas oryzae pv. oryzae (Xoo) strain PX071 in the susceptible IR24 and resistant IRBB21 rice cultivars. All extracted differential gene expression data for the rice SDRLK genes under biotic stress conditions are available in Table S4.

We extracted gene expression data for SDRLK family genes under abiotic stress conditions from E-MTAB-5941 (Buti et al., 2018), E-GEOD-38023 (Zhang et al., 2012), E-GEOD-41647, E-MTAB-4994, E-MTAB-7317 and E-MTAB-3230 (Lima et al., 2015) datasets available at the EMBL-EBI Expression Atlas (processed similarly for differential gene expression and cutoff values log2-fold change 1.0 and adjusted p ≤ 0.05) and have been analyzed by the iRAP pipeline. In addition, we extracted expression data from the Supplemental Tables associated with two peer-reviewed published articles (Hussain et al., 2016; Wu & Yang, 2020). All extracted differential gene expression data for the rice SDRLK family genes in response to abiotic stresses are available in Table S5.

The gene expression data were visualized using Python scripts (available at https://github.com/naithanis/Naithani-lab-codes). The multicluster-SN_DK.py was used to generate gene cluster-based heatmaps for baseline tissue expression data. The SN_DK_DiffX_cluster.py was used to visualize differential gene expression under biotic and abiotic stresses. The Venn diagrams were created online at http://bioinformatics.psb.ugent.be/cgi-bin/liste/Venn/calculate_venn.htpl.

Results

SDRLKs family members have the variable gene and protein structures

S-locus receptor kinase gene, the prototype member of the SDRLK gene family, consists of seven exons separated by introns: the first exon encodes the entire extracellular S-domain, the second exon encodes the transmembrane and juxta-membrane domains, and exons 3–7 encode the kinase domain (Stein et al., 1991). Previously, the SDRLK family genes that have a conserved exon-intron structure like SRK were grouped as SD1 type and genes having variable gene structures (some genes containing three introns, others containing one intron, and still others lacking introns) were grouped as the SD-2 and SD-3 types (Shiu & Bleecker, 2003; Shiu et al., 2004). The 144 rice SDRLK family genes include 53 SD1-type and 91 SD2-type genes. Out of 53 SD1 type genes, 49 contain one or more introns, and four lack introns. Out of 91 SD2 type genes, 77 lack introns, and 18 contain 1–3 introns (see Table S1).

Previously, based on homology modeling, we have suggested that the S-domain of the SRK6 protein from Brassica oleracea consists of an N-terminal signal peptide (SP) and four structurally conserved domains, including two contiguous N-terminal Lectin-like domains (LLD) followed by an EGF-like domain and a C-terminal PAN_APPLE domain (Naithani et al., 2007). Recently, the crystal structure of SRK9 bound to its cognate ligand the SCR9 confirmed this domain structure of SRK and further suggested that both N-terminal LLDs are globular and each consists of a nine-stranded β-barrel that forms a “Y”-shaped structure, the EGF-like domain contains an embedded short α-helix, and the PAN_APPLE domain is highly similar to the N-terminal domain of hepatocyte growth factor (HGF) (Ma et al., 2016). Thus, the canonical structure for SRK includes SP, LLD1, LLD2, EGF-like, PAN_APPLE, TM and Ser/Thr kinase domains (see Fig. 1A). The UniProt database provides the basic prediction for all conserved domains and SLG sequence motif contained within LLD2 for SRK except for the EGF-like domain (Fig. 1B). Though prediction for highly polymorphic EGF-like domain is not consistent across various alleles of SRK, this region has six conserved cysteine residues in all SRKs and most SDRLK genes encoding full-length RLKs.

Figure 1 Variations in the domain structure of SDRLK family proteins.

(A) Canonical domain structure of the SDRLK gene family’s prototypic member, the S-locus receptor kinase SRK based on previously published data (Naithani et al., 2007; Ma et al., 2016). (B) Predicted domain structure for SRK2-b allele of Brassica oleracea and (C) domain structure of the 144 rice SDRLK family proteins based on information available in the UniProt. Bulb-Type Lectin-like domain (LLD), EGF-like, PAN/HGF, Transmembrane (TM) and Ser/Thr Kinase domains. The “S-locus glycoprotein” (SLG) sequence motif is shared among various SRK alleles and SRK family genes.

For comparison among SDRLK family proteins, we collected protein structural domain predictions from the UniProt protein knowledgebase (see Fig. 1C). Overall, this approach provides an approximate, but useful information on the structural comparisons among the SDRLK family members. If the structural information for individual proteins/gene models is available from published studies that was given a precedence over the predictions. As shown in Fig. 1C, the SDRLK family proteins vary significantly in length, consisting of 67 full-length RLKs, 15 truncated RLKs, 14 S-domain (SD) kinases lacking a TM domain, 12 Receptor-like Proteins (RLPs) lacking a kinase domain and 36 kinases lacking any parts of the S-domain (including 10 kinases which have acquired one or more additional GnK2 domains). The detailed summary of predicted protein domains associated with the 144 rice SDRLK genes is provided in Table S1.

50% of SDRLK family genes are tandem duplicates

In the rice (O. sativa japonica Nipponbare) genome, most SDRLK family genes are located on chromosomes 1 and 4 and 82 out of 144 SDRLK family genes are located alongside or nearby to other family members (Fig. S1). Considering variable structure and difference in the gene length, we built a sequence similarity tree to identify closely related genes (see Fig. S2). Furthermore, we found that 72 (50%) SDRLK family genes that are highly similar to each other also share the same genomic neighborhood (see Figs. S1 and S2), and thus qualified as being tandem duplicates. Our criteria for selecting tandem duplicates (<5 genes in between) are more restrictive than the criteria used in the other study (Defoort, Van de Peer & Carretero-Paulet, 2019), which consider genes found in the same genomic region within a maximum of 30 genes apart as clusters of tandemly arranged genes. Our focus on a small number of genes allowed us to conduct a more detailed evaluation of gene duplication events. Overall, our analysis suggests that tandem gene duplication events have contributed significantly to the expansion and diversification of the rice SDRLK gene subfamily.

Tissue-specific expression of SDRLK genes

We found baseline tissue-specific expression data for 112 rice (O. sativa ssp. japonica) SDRLK genes in four datasets: E-GEOD-50777 (Anderson et al., 2013) contains data from the plant egg cell, pollen sperm cell, and microgametophyte vegetative cell; E-MTAB-4123 (Zhang et al., 2014) has data from the shoot, anther, carpel, and seed; E-MTAB-2037 (Sakai et al., 2011) contains data from the seed, callus, root, shoot, leaf, pre-flowering panicle, and post-flowering panicle; and E-MTAB-2039 (Davidson et al., 2012) contains data from the leaf, early inflorescence, emerging inflorescence, anther, pistil, 5 day old seed, 10 days old seed, embryo, and endosperm (see Table S2). We gathered additional information on expression for 15 additional rice SDRLK family members (see Table S3) from published literature (Aya et al., 2011; Chen et al., 2013, 2006; Fan et al., 2018; Gao & Xue, 2012; Li et al., 2018; Lin et al., 2019; Pan et al., 2020; Zhai et al., 2013; Zou et al., 2015). The SDRLK family genes show varied expression profiles: 25 genes show a unique tissue-specific expression profile, 44 genes show expression in more than one tissues, and 58 ubiquitously expressed genes show preferential expression in one or more tissues (see Fig. 2; Table S6). Overall, 86 genes express in the leaf, 83 genes express in the root, 103 genes express in the flower, 87 express in the seed and 82 express in the shoot (Fig. S3). We found expression data for eight genes in seedlings or whole plants in the differential gene expression datasets used in this study and no information for nine genes (see Table S6).

Figure 2 The rice SDRLK family genes differ in their tissue-specific expression.

The heat map of the expression data was generated based on the hierarchical clustering of genes showing similar expression patterns. Datasets: E-MTAB-2039, E-MTAB-2037, E-MTAB-4123 and E-GEOD-50777 from EMBL-EBI Gene Expression Atlas.

SDRLK family genes are regulated in response to biotic stresses

We analyzed the changes in gene expression of rice SDRLK family members in response to bacterial, viral, and fungal pathogens using publicly available transcriptomic datasets. A summary of DEG in response to various biotic agent is provided in the Table S7. Details about datasets used in this study are described in “Methods” section.

Bacterial blight response

We found two micro-array datasets E-GEOD-61833 (Girija et al., 2017), and E-GEOD-36272 in the EMBL-EBI Expression Atlas, and one published study (Narsai et al., 2013) containing expression data (see Table S4) for SDRLK family members in response to bacterial pathogen Xoo (the causal agent of bacterial blight disease in rice).

E-GEOD-61833 contains differential expression data in response to three Xoo strains: BXO43 (wildtype), BXO1002 exopolysaccaride deficient (EPS−) and BXO1003 (LPS−, EPS−) double mutant deficient in both EPS and lipopolysaccharide (LPS) (Girija et al., 2017). LPS is a component of the bacterial cell wall. Following infection, LPS is recognized by pattern recognition receptors located in the plasma membrane of the rice cells as “Microbe Associated Molecular Patterns” that in turn, activates plant triggered immunity (PTI). PTI, a broad-spectrum defense response, provides basal resistance. To counter host PTI, bacterial pathogens rely on virulent factors and the type III secretion system (T3SS). The T3SS secretes multiple effector proteins into host cells including EPS to modulate the host’s gene expression and promotes infection (Girija et al., 2017). This dataset provides an insight into interactions between counteractive mechanisms, immunity, and infection. As shown in the Fig. 3A, 10 genes, including Os01g0668901, Os01g0783800 (SDS2), Os04g0103700, Os04g0356600, Os04g0419900, Os04g0631800, Os05g0165900, Os07g0534500, Os09g0454900 and Os12g0527700 show higher expression in strain BXO1002 (LPS+ EPS−) compared to both wild-type (LPS+ EPS+) and double mutant BXO1003 (LPS−, EPS−) suggesting their role in LPS-induced and EPS-suppressed PTI. In contrast, three genes, Os01g0669100, Os01g0670100 and Os10g0101000, show higher expression in both BXO1002 (LPS+ EPS−) and BXO1003 (LPS−, EPS−) compared to BXO43, and thus, likely to be independent of LPS induced PTI, but down-regulated by EPS-mediated signaling. Os02g0472700 and Os04g0226600 express at similar levels in all three strains and are likely to be independent of LPS and EPS signaling.

Figure 3 SDRLK genes are regulated in response to bacterial pathogen X. oryzae.

(A) Differential gene expression in rice leaves infected with three X. oryzae pv. oryzae (Xoo) strains, BXO43 (wild-type), BXO1002 (EPS−), and BXO1003 (LPS−, EPS−). (B) Differential expression of genes in two susceptible rice varieties IR24 and Nipponbare in response to various X. oryzae strains. (C) Changes in gene expression in response to virulent Xoo strain PX071 in a susceptible IR24 and a resistant IRRBB21 genotypes after 24 and 96 h post-infection.

The E-GEOD-36272 contains differential expression data for 41 SDRLK family members from susceptible rice varieties IR24 (O. sativa indica IR24) and Nipponbare (O. sativa japonica Nipponbare) in response to nine X. oryzae strains. The X. oryzae pv. oryzicola (Xoc) strain BLS303 causes bacterial streak diseases and not blight. The wildtype X. oryzae pv. oryzae (Xoo) strains T7174, PX086, PX099A-2 and PX099A are pathogenic and cause blight disease. PXO99AME7 is a non-pathogenic strain (lacking a functional T3SS), and PXO99AME5 (unknown mutation), PXO99AME2 (pthXo1), PXO99AME1 (pthXo6 and avrXa27) are low virulent strains containing mutations in the transcription activator-like (TAL) genes. The pthXo1 is a major TAL effector gene in the PX099A that promotes disease in rice (host) cells. AvrXa27, another effector from Xoo activates transcription of a disease-resistant gene Xa27 in rice. The loss of pthXo1 (in PXO99AME2) or pthXo6, and avrXa27 (in PXO99AME1) results in low virulence. This dataset can be used for identifying genes specifically regulated in response to PXO99A in susceptible rice varieties IR24 and Nipponbare. Figure 3B shows the expression profile of 41 genes in IR24 and Nipponbare. Interestingly, Os01g0668901 and Os10g0342300, both encoding full-length RLKs, are upregulated in IR24 and Nipponbare in response to Xoo wildtype strains, but not in response to T3SS defective mutant or TAL mutants suggesting their role in pthXo1 mediated effector-triggered immunity (ETI). Os01g0668901 is highly expressed in reproductive cells (pollen sperm cell, egg cell, and embryo). However, Os10g0342300 shows preferential expression in root and flower but also expresses in leaf, shoot and seeds (Aya et al., 2011). Os05g0166300, Os05g0165900 and Os01g0669100 are generally upregulated in response Xoo in Nipponbare, but seems to be independent of the T3SS derived effectors (likely to be part of PTI). Os04g0419700 shows up-regulation only in the IR24 genotype in response to PXO99AME1 and PXO99AME2, suggesting its down-regulation by pthXo1. Os06g0165500 and Os08g0179000 are likely to be down-regulated by pthXo1 in IR24 in response to PX099A. Os04g0420033 is specifically down-regulated in Nipponbare in response to all Xoo strains and likely to be up-regulated by avrXa27. In response to Xoc strain BLS303, five genes (Os01g0668600, Os01g0670100, Os02g0710500, Os04g0632600 and Os06g0602500) in IR24 and seven genes (Os01g0890600, Os04g0226600, Os04g0631800, Os04g0632600, Os05g0165900, Os07g0534500, Os10g0136400 and Os10g0136500) in Nipponbare show differential expression, however, only Os04g0632600 is common in both cultivars.

The third dataset, extracted from Narsai et al. (2013), contains expression data for 14 SDRLK family genes from a susceptible rice cultivar IR24, and a resistant cultivar IRBB21 in response to Xoo strain PX071 infection. As shown in Fig. 3C, 10 genes are up-regulated in resistant IRBB21 (but not in susceptible IR24) in response to PXO71: one gene Os10g0136400 shows up-regulation during 24 h post-infection (early), five genes (Os03g0838100, Os04g0632600, Os06g0689600, Os07g0534500 and Os10g0136500) show up-regulation during 96 h post-infection (late), and four genes (Os04g0226600, Os04g0655300, Os08g0343000 and Os06g0602500) show up-regulation early on and their expression is maintained in the late phase. Two genes Os01g0784700 and Os07g0551300 are down-regulated only in resistant IRBB21during the late phase. Also, Os01g0670300 is down-regulated and Os12g0527700 is up-regulated only in the susceptible IR24.

In the E-GEOD-36272, we found data for 41 SDRLK family members that includes 16 out of 18 genes in E-GEOD-61833. The missing two genes Os07g0550500 and Os10g0101000 from E-GEOD-61833 show expression only in response to the EPS− strains. Further, eight out of 14 genes from Narsai et al. (2013) are common with E-GEOD-36272 and remaining six genes (Os01g0670300, Os03g0838100, Os04g0655300, Os06g0689600, Os07g0551300 and Os08g0343000) are unique. Total 49 rice SDRLK family genes show differential expression in response to Xoo. The presence of common DEG across three independent studies strongly suggests their role in Xoo pathogenesis and plant immunity.

Bacterial panicle blight response

Burkholderia glumae causes bacterial panicle blight disease of rice that is characterized by having florets with a darker base and a reddish-brown margin, and frequently upright due to poor filling (Magbanua et al., 2014; Nandakumar et al., 2009). This disease also causes sheath rot and seedling blight. E-MTAB-4406 RNA-Seq dataset (Magbanua et al., 2014) contains data for 72 SDRLKs family genes (see Table S4) that was generated from seedlings and inflorescence of a moderately resistant rice genotype CL 161 and a susceptible rice genotype CL 151 infected with B. glumae pathogenic strain 89gr-4. Figure 4 shows 18 DEG when resistant CL 161 genotype is compared with the susceptible CL 151 genotype in response to B. glumae infection. Five genes (Os01g0222800, Os03g0221700, Os04g0201900, Os04g0633200 and Os12g0130200) show up-regulation and two genes (Os03g0556600 and Os04g0475200) show down-regulation at the flowering stage in CL 161 in comparison to CL 151. Three genes (Os01g0889900, Os06g0689600 and Os11g0133001) show up-regulation at seedling as well as the flowering stage in CL 161 in comparison to CL 151. Six genes (Os01g0113350, Os01g0155200, Os01g0223900, Os01g0668600, Os09g0551400 and Os11g0441900) are up-regulated in CL 161 compared to CL 151 at the seedlings stage. Os07g0534700 and Os07g0550900 are down-regulated at the seedling stage but up-regulated at flowering stage in CL 161 compared to CL 151.

Figure 4 Differential expression of SDRLK family genes in response to bacterial pathogen B. glumae.

The data is from the seedlings and inflorescence of a moderately resistant (R) CL 161, and a susceptible (S) CL 151 rice genotypes after 48 h post-inoculation with B. glumae.

Stripe disease response

Rice stripe virus (RSV), a single-stranded RNA virus, causes stripe disease in rice. This virus is transmitted through brown plant hoppers Laodelphax striatellus only, and direct plant to plant transmission of the virus doesn’t occur. E-MTAB-5025 dataset contains expression data from rice plants infected using viruliferous insect vector L. striatellus or by microinjection containing insect-/plant-derived RSV(Zhao et al., 2016). It was demonstrated that plants inoculated with plant-derived RSV using microinjection failed to cause stripe symptoms, whereas plants directly infected by insect vectors or microinjected with insect-derived RSV showed symptoms of the disease suggesting that the RSV acquires pathogenicity during its replication inside the insect vector (Zhao et al., 2016). We found expression data for 41 SDRLK family genes in E-MTAB-5025 (see Table S4). As shown in Fig. 5A, no significant change in gene expression was observed when plant-derived RSV was microinjected into healthy plants. In contrast, plants infected with insect-derived RSV (microinjected or delivered via viruliferous insect vector) share five down-regulated genes (Os01g0890600, Os11g0133100, Os12g0130200, Os12g0130300 and Os12g0527700) and seven up-regulated genes (Os01g0224000, Os02g0710500, Os03g0556600, Os04g0103500, Os04g0420900, Os07g0550900 and Os09g0454900). This is consistent with the previous observation that in both categories of plants show typical stripe disease symptoms and damage to chloroplast structure (Zhao et al., 2016). Further experimental analysis will be required for deciphering the role of these twelve genes in rice stripe disease development/resistance.

Figure 5 Rice SD-RLK family genes are regulated in response to Rice Stripe Virus (RSV) and fungal pathogen R. solani.

(A). Change in the expression of SDRLK family genes in rice leaves that were mechanically inoculated with insect-derived RSV and plant-derived RSV in comparison to the rice leaves fed on by the natural viral vector viruliferous L. striatellus. (B) Differential expression of the rice SDRLK family genes in response to fungal pathogen R. Solani in a susceptible rice variety Cocodrie (CCDR) and a resistant rice variety MCR010277 (MCR) after 3 and 5 days post-infection (DPI).

Leaf-sheath blight response

The fungal pathogen Rhizoctonia solani causes leaf sheath blight disease of rice. E-MTAB-6402 (ERS2106879–ERS2106902) dataset contains expression data for 39 SDRLK family genes (see Table S4) from a resistant rice genotype MCR010277 (MCR), and a susceptible rice Cocodrie (CCDR) infected with R. solani strain LR172 for 3 days (early) and 5 days (late). Both CCDR and MCR are long-grain, high-yielding breeding line developed in the USA (https://www.lsuagcenter.com). As shown in Fig. 5B, six genes (Os01g0668901, Os01g0784700, Os01g0889900, Os02g0472700, Os02g0710500 and Os07g0534700) show up-regulation and four genes (Os01g0668400, Os04g0201900, Os04g0631800 and Os04g0632901) show down-regulation in resistant MCR at 3 Days Post Infection (DPI) but not in susceptible CCDR. Os07g0550500 shows up-regulation at 3 DPI as well as at 5 DPI in the resistant MCR but not in CCDR. No change in gene expression is observed in CCDR 5-days post-infection. Three genes Os01g0670100, Os01g0890600, and Os06g0689600 show increased expression in CCDR at 3-days post-infection compared to MCR. In total, 14 rice SDRLK family genes show differential expression in response to R. solani.

SDRLKs are regulated in response to abiotic stresses

We analyzed the changes in gene expression of rice SDRLK family members in response to various abiotic stresses using publicly available transcriptomic datasets (see Table S5 and “Methods” for details). A summary of genes involved in one or more abiotic stress response is provided in Table S7.

Chilling response

Rice is sensitive to chilling stress, especially at the seedling stage, which negatively impacts the plant’s growth and yield. To understand SDRLK family members’ role in chilling/cold tolerance and susceptibility, we analyzed two datasets E-MTAB-5941 (Buti et al., 2018) and E-GEOD-38023 (Zhang et al., 2012). We find 37 genes common to both datasets; 18 unique in E-MTAB-5941; and 22 unique in the E-GEOD-38023 (total 77 genes).

E-MTAB-5941 RNA-seq dataset provides an opportunity to compare the short- and long-term stress-induced changes in the transcriptome of a chilling-sensitive genotype Thaibonnet and a chilling-tolerant genotype Volano, each subjected to 2 and 10 h chilling treatment at 10 °C (Buti et al., 2018). As shown in Fig. 6A, overall, 55 SDRLK genes show differential expression in response to chilling stress in the two genotypes. In comparison to Thaibonnet, the Volano shows up-regulation of four genes (Os05g0166300, Os04g0419900, Os06g0164900 and Os04g0632100) in response to short-term chilling stress; up-regulation of four genes (Os07g0534500, Os01g0669100, Os06g0602500 and Os07g0534700) in response to long-term chilling stress; and up-regulation of Os07g0535800 in both short- and long-term chilling stress. Also, six genes (Os10g0136500, Os06g0541600, Os10g0136400, Os04g0633300, Os01g0890600 and Os12g0527700) show decreased expression in the tolerant Volano compared to the susceptible Thaibonnet. Overall, 15 DEGs from this subfamily are likely to play a role in the chilling tolerance gene-networks.

Figure 6 Rice SDRLK family genes show differential expression in response to chilling.

(A) Fold change in the expression of 55 genes in susceptible Thaibonnet and tolerant Volano genotypes treated with chilling stress (10 °C) for 2 and 10 h (data source: E-MTAB-5941). (B) Differential expression of 59 genes in susceptible IR29 and tolerant LTH genotypes in response to chilling stress at 4 °C for 2, 6, 24 and 48 h as well as after 24 h recovery from 24 h stress (data source: E-GEOD-38023).

E-GEOD-38023 contains expression data from a chilling-tolerant Li-Jiang-Xin-Tuan-He-Gu (LTH) japonica landrace variety and chilling-sensitive IR29 indica cultivar. The plants from both genotypes were subjected to chilling treatment at 4 °C for 2, 8, 24, and 48 h, and then moved to normal temperature 29 °C for 24 h to allow recovery (Zhang et al., 2012). We found expression data for 59 SDRLK genes in this dataset (see Table S5). As shown in Fig. 6B, under chilling stress, many common genes are up-regulated in both genotypes at an early stage. But, at later stages, we find up-regulation of genes in the chilling-tolerant LTH and strong repression in the chilling-sensitive IR29. Specifically, in LTH genotype, eight genes show up-regulation (early response genes Os01g0224000, Os10g0101000; and late response genes Os04g0632600, Os03g0556600, Os12g0640700, Os01g0155200, Os07g0534700, Os04g0420900), and three genes (Os05g0166300, Os01g0889900 and Os01g0223900) show down-regulation in comparison to IR29. Also, five genes (Os04g0419900, Os06g0602500, Os03g0422800, Os06g0690200 and Os01g0885700) show increased expression in LTH in response to chilling at early stage and their expression is maintained in the late phase (Fig. 6B). Overall, 16 SDRLK family genes are differentially regulated in response to chilling between the tolerant and the susceptible rice genotypes.

Drought response

We found drought-induced differential gene expression data for 59 SDRLK family genes from three microarray datasets (E-GEOD-41647, E-MTAB-4994 and E-MTAB-3230), and one RNA-seq dataset E-MTAB-7317 (see Table S5). The data from four datasets were compiled together for visualization (direct comparison across datasets were not made, but a summary is given in Table S7).

The E-GEOD-41647 contains data for 26 SDRLK family genes from seedlings of susceptible IR20 and drought-tolerant Dagad deshi genotypes. We find five genes (Os01g0223700, Os01g0669100, Os03g0422800, Os06g0575400 and Os08g0179000) are down-regulated and 10 genes (Os04g0420900, Os04g0632100, Os09g0454900, Os04g0632600, Os06g0689600, Os06g0690200, Os06g0241100, Os06g0602500, Os01g0224000 and Os01g0885700) are up-regulated in the Dagad deshi but not in the IR24 (see Fig. 7). Thus, 15/26 genes from this set appear are likely to play a role in the drought response pathways. The E-MTAB-4994 contains data for 20 SDRLK subfamily genes from the flag leaf at the panicle initiation stage of Oryza sativa Indica cultivar Nagina 22 (a drought-tolerant genotype). As shown in Fig. 7, 10 genes show up-regulation in response to drought in Nagina 22. E-MTAB-3230 contains data for eight genes from seedlings of Nagina 22 and an enhanced drought-tolerant mutant ewst1 of Nagina 22 grown in hydroponic culture subjected to 25% polyethylene glycol (PEG) for one hour (Lima et al., 2015) to mimic drought stress. We find that six genes are down-regulated and two genes are up-regulated in ewst1 mutant compared to Nagina 22 in response to PEG treatment. The two up-regulated genes Os07g0550900 and Os07g0551300 are unique to the ewst1 mutant (see Fig. 7).

Figure 7 Rice SDRLK family genes show drought-induced differential gene expression.

Data from four datasets E-GEOD-41647, E-MTAB-4994, E-MTAB-3230 and E-MTAB-6402 was used for comparison across susceptible (S) and tolerant (T) rice genotypes.

E-MTAB-7317 contains data for 44 SDRLK family genes from drought tolerant Nagina 22 and a salt-tolerant genotype Nonabokra (drought tolerance unknown), each subjected to drought for 2-days (early) and 3-days (late). As shown in Fig. 7, many SDRLK family members show similar expression patterns between Nagina 22 and Nonabokra, suggesting that the latter might possess some level of drought tolerance. However, in comparison to Nonabokra, Nagina 22 shows a significant difference in the drought response: eight genes (Os01g0223800, Os01g0670100, Os01g0890600, Os04g0420033, Os05g0166300, Os07g0534500, Os07g0550500 and Os12g0527700) are down-regulated, and seven genes (Os01g0222800, Os01g0366300, Os04g0632100, Os04g0633600, Os04g0634400, Os12g0130300 and Os12g0130800) are up-regulated in Nagina 22 but not in Nonabokra. Also, Os06g0165500 is up-regulated early in Nagina 22, whereas in Nonabokra it is up-regulated during the late phase.

Overall, three drought-tolerant genotypes, Dagad deshi, Nagina 22, and ewst1_Nagina 22 share four common up-regulated genes (Os01g0224000, Os04g0632100, Os06g0690200 and Os09g0454900) and four common down-regulated genes (Os01g0223700, Os01g0669100, Os03g0422800 and Os08g0179000) in response to drought suggesting their conserved role in drought response.

Submergence response

Rice (Oryza sativa) is unique among cereals that can germinate under submerged conditions and tolerate an anoxic environment. The ability to withstand short-and long-term submergence stress varies widely among the rice cultivars and is crucial to their adaptation to a range of geographical regions, particularly the world’s coastal regions. Considering RLKs play roles in cell signaling, we searched for gene expression data for SDRLK family genes under submergence stress and found data for 35 genes in the Supplemental Tables associated with two recent publications (Hussain et al., 2016; Wu & Yang, 2020) (see Table S5).

Wu & Yang (2020) investigated the role of auxin signaling in the coleoptile growth of Oryza sativa L. japonica cultivar Taikeng-9 seedlings under submergence and showed that a polar auxin transport inhibitor 2,3,5-triiodobenzoic acid (TIBA) caused delay in seeds germination and reduced growth of the coleoptile under submergence. Furthermore, the authors identified 3,448 up-regulated and 4,360 down-regulated genes in response to submergence and TIBA treatment. We found gene expression data for 20 SDRLK family genes in this dataset. As shown in Fig. 8, five genes (Os04g0103700, Os07g0535800, Os02g0710500, Os01g0587400 and Os05g0166300) show down-regulation, and fifteen genes show up-regulation in the submerged seedling supplied with TIBA in comparison to submerged seedling (without TIBA). These results suggest the association of these SDRLK genes with submergence response and auxin-mediated signaling required for coleoptile growth.

Figure 8 Rice SDRLK family genes are regulated in response to submergence.

Expression data for SDRLK family genes from Wu & Yang (2020) and Hussain et al. (2016) were plotted side by side for comparison between india and japonica rice cultivars under submergence.

Hussain et al. (2016) studied the protective effects of seed priming under submergence in Indica inbred Oryza sativa L. cultivar Huanghuazhan by comparing transcriptomes of 4-day-old seedlings. The samples included submerged seedlings grown from non-primed and primed seeds with selenium (Se) or salicylic acid (SA). In this dataset, we found nineteen SDRLK family genes showing differential expression (see Table S5). The Se-priming of rice seeds prior to submergence modified the expression of four genes (Os06g0496800, Os01g0669100, Os03g0221700 and Os04g0419900) when compared to non-primed (NP) submerged plants, and mimic the control plants not subjected to submergence stress. Similarly, the SA-priming modified the expression pattern of Os06g0241100, Os07g0141100, Os02g0767400 and Os06g0496800 in comparison to and non-primed (NP) submerged plants (see Fig. 8). In both, SA- and Se-primed seedlings Os06g0496800 is up-regulated to mitigate the effect of submergence. In addition, Os12g0130300 shows increased expression in SA- and Se-primed seedlings but no differential regulation in response to submergence, suggesting that this gene may be affected by the materials or methods used for seed priming. Notably, four genes Os06g0241100, Os06g0575000, Os07g0141100 and Os05g0166300 up-regulated under submergence in this dataset are in common with the dataset from Wu & Yang (2020), suggesting their conserved function in both rice cultivars.

Common genes regulated by abiotic and biotic stresses

In total, 115 SDRLK family genes show differential expression in response to environmental stimuli, including eight genes not expressed under normal growth conditions. Specifically, 36 genes are significantly regulated under abiotic stress conditions in tolerant genotypes in comparison to susceptible genotypes; 17 genes are significantly regulated under biotic stresses in resistant genotypes in comparison to susceptible genotypes; and 34 genes regulated in response to both biotic and abiotic stress conditions between tolerant and susceptible genotypes (see Table S7). We find 81 common genes regulated in response to both biotic and abiotic stress conditions also express under normal growth and developmental conditions (Fig. S3). Notably, a few genes expressed in the seed under normal growth conditions are highly expressed in leaf and other vegetative tissue under stress. For example, Os05g0501400 expresses only in seed under normal growth condition, but expresses in seedlings under submergence; Os09g0551400 shows low expression in seed but up-regulated seedlings of resistant genotype CL 161 in response to B. glumae infection. This is consistent with the previous observation that many genes involved in the seed development are also regulated under stress (Naithani, Nonogaki & Jaiswal, 2017).

Discussion

The rice SDRLK subfamily of RLKs includes 144 members, including full-length gene duplicates, partial gene duplicates, and pseudogenes. A total of 72 (50%) genes of the rice SDRLK family are tandem duplicates, and the majority is in chromosomes 1 and 4. In general, the expansion and diversification of the SDRLK gene family in rice results from the tandem gene duplication of full-length or partial gene fragments, recombination, and ancient whole-genome duplications. This is consistent with this family’s evolution in other plant species (Lehti-Shiu et al., 2009; Myburg et al., 2014; Shiu & Bleecker, 2003). This gene subfamily was named after SRK, the female determinant of SI response in the Brassica and most of its members (full-length RLKs) bear gene annotation “S-locus Receptor-like Kinase” involved in “SI response”. These gene annotations were assigned by automated GO pipelines currently being used for large-scale gene annotation of sequenced genomes. Other genes of the family that are partial gene duplicates are assigned “unknown function” (see Table S1). The automated projections of gene annotations from the well-characterized orthologs in other organisms is a powerful approach for identifying highly conserved genes and for preliminary gene annotations at the genome-scale. However, it falls short in assigning correct gene annotations and functional associations to the individual members of large gene families. Often, most members of a gene family or its subclass bear similar gene annotation without considering the evolutionary fate of gene duplicates, such as sub-functionalization, functional diversification, neo-functionalization, and pseudogenization. Furthermore, rice (O. sativa) is a self-fertilizing and the sporophytic SI does not exist in this species. Considering SDRLK gene family’s size, inaccurate gene-annotations and functional associations impacts all types of large-scale genomic and transcriptomic analyses. Thus, a manual biocuration approach is needed for correcting gene annotations of the SDRLK gene family.

To date, the functions of most SDRLKs are unknown, however, it is increasingly becoming clear that these genes are expressed in a variety of tissues and in response to both biotic and abiotic stresses (Lehti-Shiu et al., 2009; Vining et al., 2015) and play crucial roles in many important plant processes beyond SI, such as in root development (Pan et al., 2020), leaf senescence (Chen et al., 2013), stem elongation (Li et al., 2018), in grain yield, and pathogen resistance in rice (Kim et al., 2009; Wang et al., 2015; Zou et al., 2015). Moreover, we have vast amount of publicly available transcriptomic data that can reveal new information about the expression of the SDRLK family genes and their associations with important biological processes. We utilized publicly available genomic and transcriptomic data to study the structure of rice SDRLK subfamily genes and their expression across various tissues under normal growth and developmental conditions as well as in response to pathogen infection and abiotic stresses; and conducted a thorough review of the published literature to gather additional information. We found expression information for 135 rice SDRLK family genes. A total of 128 rice SDRLK family genes are expressed under normal growth and developmental conditions, including 27 genes showing unique tissue-specific expression, 43 genes showing preferential expression in more than one tissue, 58 genes showing ubiquitous expression but varying in degree of expression across tissue types (see Fig. 3; Table S6). Furthermore, 36 SDRLK genes that are expressed under normal growth are not regulated in response to any stress condition. However, tissue-specific expression patterns suggest their potential role in plant development and/or formation of specific organs/tissues/cell types. One gene from this group, PSRK2, controls stem elongation by negatively regulating GA responses (Li et al., 2018).

Overall, 99 SDRLK family genes show differential expression in response to environmental stimuli, including seven genes not expressed under normal growth conditions (Table S7). Specifically, 39 genes are regulated under abiotic stress conditions; 23 genes regulated under biotic stresses; and 37 genes regulated in response to both biotic and abiotic stress conditions. In general, we find that most SDRLK family genes respond to more than one stress condition, which seems to be consistent with the role of many kinases in cell-signaling. In general, we find overlapping expression patterns in many tandem duplicates and identical tissue-specific expression profiles for 10 pairs of tandem duplicates (highlighted in bold in Table S6) and identical profiles of four pairs in response to stress treatment (highlighted in bold in Table S7). Notably, Os08g0179000 shows flower-specific, and Os08g0179150 shows preferential expression in flower; however, both genes are predicted to encode two complementary halves of an RLK, and correspond to one gene LOC_Os08g08140 in the MSU database. Gene models for both genes are likely incorrect in the RAP database, and a correction is needed.

Based on a careful evaluation, we have created a list of candidate genes that are likely to play an important role in conferring tolerance to one or more abiotic stresses and/or resistance to pathogens (Table S7). A few genes from this family have been experimentally characterized in detail and their role in conferring biotic resistance and abiotic stress tolerance has been confirmed. In particular, Os06g0494100 (PID2) confers race-specific resistance to M. grisea (Wang et al., 2015); Os01g0783800 (SDS2) controls programmed cell death and provides resistance to Magnaporthe oryzae (Fan et al., 2018); Os01g0223800 (ESG1), induced by treatment with PEG, NaCl, abscisic acid (ABA) and drought, provides drought tolerance (Pan et al., 2020); and Os07g0186200 (SIK2), also induced by NaCl, drought, cold, dark, and ABA, is involved in drought and salt tolerance, and delays dark-Induced leaf senescence (Chen et al., 2013). Also, Os01g0669100 (LSK1) controls branching per panicle and grain yield (Zou et al., 2015). We expect that many genes listed in Table S7 are likely to play a role in mitigating one or more stresses and can be exploited for breeding advanced high-yielding rice varieties. Some SDRLK family genes have the potential for direct applications in improving and managing crop productivity. For instance, the submergence tolerance traits could be exploited commercially to manage weeds in paddy fields, where controlled flooding would allow the germination of rice seeds and seedlings’ growth but would suppress the weeds. However, further functional assays and phenotypic studies on cultivated and wild rice species will be required on these candidate genes to ascertain their exact function.

We aim to improve and enrich the gene description in public databases and help the plant research community identify potential candidate genes. Here, we re-used and re-analyzed publicly data to synthesize new information about the structure and function of SDRLK gene family members. Based on the analysis of gene structure, transcription profile, and extensive literature review, we propose the gene nomenclature and gene description for the 144 members of the rice SDRLK family (see Table S1). We will integrate this information in the Plant Reactome knowledgebase (Naithani et al., 2020) that also exchanges information with other public resources such as UniProt and the RAP database.

In addition to retrieving new knowledge and enriching public databases, manual biocuration projects teach students/researchers genomic data evaluation, help to build the community of biocurators (Naithani et al., 2019), and provide peer-review of the public genomic datasets. The re-use of genomic and transcriptomic data for mining new insights (“data parasitism”) is already making significant impact in the field of genomics and the value of secondary data analyses and meta-analyses is being recognized. The data re-use is one of the core component of FAIR data principle. We expect that our study sets forth an example of re-use and re-analysis of genomic data for improving gene annotation of the large gene families and enriching the contents of public genomic databases and resources via community biocuration efforts.

Conclusions

Plant genomes harbor extensive gene duplications and large gene families consisting of members ranging from redundant gene duplicates, gene duplicates showing tissue- and developmental stage-specific function, genes expressed only under stress conditions, genes with new/diversified functions, and pseudogenes. However, the individual members of the gene families are poorly annotated in the public databases. Thus, researchers often ignore these genes as potential candidate genes for further experimental analysis. We utilized publicly available genomic and transcriptomic data to improve gene annotation and descriptions of rice SDRLK subfamily genes and identified potential candidate genes likely to play a role in conferring tolerance to one or more biotic and abiotic stresses. We see tremendous potential in re-analysis of publicly available genomic data for improving the annotations of the members of large gene families, building gene-gene interaction networks, and identifying candidate genes likely to impart important traits related to yield and climate resilience. Manual biocuration of genes by researchers can improve the contents of genomic databases and complement high-throughput automated protocols.

Supplemental Information

Supplemental Information 1 50% rice SDRLK family members are tandem gene duplicates.

(A) Distribution of 114 SDRLK family genes on 12 rice chromosomes. (B) A close view of tandemly arranged genes on chromosome 4 showing Os04g0631800 (OsRLCK162), Os04g0632100 (OsRLCK163), Os04g0632500, Os04g0632600, and Os04g0632901 genes.

Click here for additional data file.

Supplemental Information 2 A sequence similarity tree of the rice SDRLK subfamily proteins.

Tips were aligned to the right-hand side, and the branch length (indicative of the evolutionary distance between the sequences) information is displayed at the top of each branch. The blue color nodes and labels in the tree represent full-length RLKs. Groups of genes shaded with the same color depict tandem duplicates.

Click here for additional data file.

Supplemental Information 3 A summary of gene expression patterns of the rice SDRLK family genes.

Click here for additional data file.

Supplemental Information 4 Propose ’gene nomenclature’ and ’gene descriptions’ for the rice SDRLK family genes based on gene/protein structure and expression profile.

Click here for additional data file.

Supplemental Information 5 A list of transcriptome datasets used in this study.

Click here for additional data file.

Supplemental Information 6 Tissue-specific expression data used in this study.

Click here for additional data file.

Supplemental Information 7 Differential expression data for SDRLK family genes in response to biotic stress.

Click here for additional data file.

Supplemental Information 8 Differential expression data for SDRRLK family genes in response to abiotic stress treatments.

Click here for additional data file.

Supplemental Information 9 A summary of tissue-specific gene expression of the rice SDRLK family genes.

Click here for additional data file.

Supplemental Information 10 A summary of changes in the expression of the rice SDRLK family genes in response to biotic and abiotic stresses.

Click here for additional data file.

We acknowledge Justin Preece (Jaiswal lab, Oregon State University) for helping DD with Python scripting.

Additional Information and Declarations

Competing Interests

Author Contributions

Data Availability

Pankaj Jaiswal is an Academic Editor for PeerJ.

Sushma Naithani conceived and designed the experiments, performed the experiments, analyzed the data, prepared figures and/or tables, authored or reviewed drafts of the paper, and approved the final draft.

Daemon Dikeman performed the experiments, analyzed the data, prepared figures and/or tables, authored or reviewed drafts of the paper, and approved the final draft.

Priyanka Garg performed the experiments, prepared figures and/or tables, and approved the final draft.

Noor Al-Bader performed the experiments, authored or reviewed drafts of the paper, and approved the final draft.

Pankaj Jaiswal performed the experiments, authored or reviewed drafts of the paper, and approved the final draft.

The following information was supplied regarding data availability:

Python scripts used for gene expression data visualization, the multicluster-SN_DK.py and SN_DK_DiffX_cluster.py are available at GitHub: https://github.com/naithanis/Naithani-lab-codes.

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
