# Peer review of "Beyond gene ontology (GO): using biocuration approach to improve the gene nomenclature and functional annotation of rice S-domain kinase subfamily"

_PeerJ, doi:10.7717/peerj.11052_

## Round 0.1 · original submission · Major Revisions

Please carefully revise the manuscript and address all the concerns pointed out from reviewers.

Reviewer 1 ·

Basic reporting

Not professional article structure, figures, tables.

Experimental design

Very little experimental design.

Validity of the findings

Impact and novelty are not enough.

Additional comments

The paper wants to present the evolutionary expansion, gene structure and expression of the rice S-domain kinase gene family. But the article structure is not professional or even a mess, especially the many figures, big and many tables with a mess, without any conclusion pattern. Moreover, there is very little experimental design in the paper with only analysis and data from websites. So the Impact and novelty is not enough.

Reviewer 2 ·

Basic reporting

Overall, the manuscript is well written.
I suggest the authors modify the following sections:
1: Line 83-93, the authors listed 7 different SDRLKs to show their roles in diverse biological processes. But there is no logic transition between them. I'd like to recommend them to condense these sentences according to the roles of SDRLKs. For example, plant defense against pathogens, plant development and so on. That would improve the readability of this paragraph.
2: Line 130, gene structures, protein domains, and subcellular localizations.
3: Line 167, biotic stresses.
4: Line 202-line 206, there is grammar mistake here, rewrite this sentence.
5: Line 243, Our criteria of selecting tandem duplicates is more restrictive compared to the criteria used in other studies.
6: Line 321, italicize pthXo6 and avrXa27.
7: Line 517, italicize ewst1 mutant.
8: Line 162-165, rewrite this sentence.

Experimental design

The experiments are well designed.
I would like to suggest the authors improve the following sections:

1: Introduce those tissue-specific expression datasets of SDRLKs. For example, E-MTAB-4123 and E-MTAb-2037 both have data from shoot. What kind of plants are they? Do they use similar growth conditions? E-MTAB-4123 and E-MTAB-2039 both have data from anther. It might be good if the authors can give some background introduction of these data.
2: In line 368, 49 SDRLK family genes were found to be deferentially regulated in response to various strains of Xoo. Is there any known published information about these genes? What are their functions in plant?
3: In the Leaf sheath blight response paragraph ( Line 424 to line 437), a resistant rice genotype 427 MCR010277 (MCR) and a susceptible rice genotype Cocodrie (CCDR) were used. What are the difference between the two genotypes?
4: In the chilling response paragraph (line 459), what is the temperature used?

Validity of the findings

The authors identified many differential expressed genes regulated by abiotic or biotic stresses. But there is not much explanation about the validity and novelty of this discovery. For example, in line 580, it says 115 SDRLK family genes show differential expression in response to environmental stimuli. There is no introduction of these 115 genes. How valid is this discovery? Are they all newly identified ones? In line 643, The authors cited that “One gene from this group, PSRK2 is known to control stem elongation 643 by negatively regulating GA responses (Li et al. 2018)”. This is great as a way of validation of the analysis. I suggest them provide more published information about them. If it is not available in ricet, information about the orthologs in Arabidopsis will also be helpful to improve the validity of the finding and the conclusion.

Additional comments

Overall, I think the authors did a great job of identifying new genes associated with biotic and abiotic stresses in rice. Experiments were well-designed and performed. Results were well-explained.

Reviewer 3 ·

Basic reporting

All comments are appended in the section "General comments for the author".

Experimental design

All comments are appended in the section "General comments for the author".

Validity of the findings

All comments are appended in the section "General comments for the author".

Additional comments

In this manuscript, the authors have carried out the mining of pubic data (genomic and transcriptomic) for exploring and reanalysis of the rice S-domain kinase gene family.
I have the following comments:
1. Authors have retrieved the data from different publicly available databases and performed their analyses. For example, authors have used the transcriptome data from various resources viz., EMBL-EBI Expression ATLAS, ArrayExpress, and literature survey. I suppose different platforms, tools, and their versions have been previously used in the analyses (original) of the data which the authors have re-used in the present study. I am curious, how do authors justify their results by just visualizing differently analyzed datasets and making their hypothesis over these results? In principle, authors could have re-analyzed the data from different public resources using a common pipeline followed by focusing their analysis on the S-domain kinase gene family. However, authors have just taken the results of previous studies and presented the specific component consisting of the S-domain kinase gene family. Please justify this.
2. The introduction section lacks the motivation and the knowledge gap which persuaded the authors for this work. Moreover, the discussion section must be in parallel to the introduction section. So, I suggest the authors revise this section accordingly.
3. Provide either a web link or the reference for a given citation, revise the manuscript thoroughly.
For example, We used the rice genome browser provided by the Gramene (http://gramene.org) (Tello-Ruiz et al. 2018) to investigate gene and transcript models of 144 SDRLK family members.
4. What are the “short genes” which are removed while pre-processing the data for the analysis? Are these the partial genes?
5. “We defined genes which share high sequence similarity, and are present on the same chromosomes alongside or have less than five genes in between as tandem duplicates”.
Define the “high sequence similarity” parameter here.
6. Have the authors used PLAZA 4.5 in the current study? If so, provide the details in methodology.
7. The conclusion section must be very concise and to the point with a clear-cut inference of the manuscript. Revise it accordingly.
8. The section “The second dataset, E-GEOD-36272 shows differential expression of 41 SDRLK family ….” has no reference. Please cite the section.
9. Correct the grammar and language throughout the manuscript.
For example:
a. … , and the kinase domain is contained on exons 3 to 7 (Stein et al. 1991).
b. …. lack introns. and out of 91 SD2 type genes 77 have no introns and 18 genes contain 1-3 introns (see Supplementary Table 1).
c. Recently, crystal structure of SRK9 along with its cognate ligand the S-locus cysteine-rich 9 (SCR9) has confirmed this domain configuration of SRK (Ma et al. 2016) and …
10. Italicize gene names throughout the manuscript.

---

## Round 0.2 · Minor Revisions

Please modify the two Tables the reviewer pointed out before final acceptance.

Reviewer 1 ·

Basic reporting

OK.

Experimental design

OK.

Validity of the findings

OK.

Additional comments

The information and results of the two Tables should be not repeated with all the figures.

Reviewer 2 ·

Basic reporting

no comment

Experimental design

no comment

Validity of the findings

no comment

---

## Round 0.3 · Major Revisions

First, let me apologize for the delay in getting back to you. The previous academic editor was not able to complete their duties handling the manuscript, so I have taken over.

The previous academic editor had sent your manuscript to the Section Editor, Gerard Lazo (who has final approval) and Gerard had a number of points that need to be addressed before acceptance. I'll summarize below and then paste in his full comments. I know that this is not what you wanted to hear at this stage, but these revisions will result in a better manuscript and make your work more accessible to the community.

My interpretation of Gerard's comments are to

1) Use your analysis to add GO or other ontology terms to the genes in your paper. Given your lab's involvement in many annotation and ontology projects this should be easy to accomplish. (In fact, given your lab's expertise we were both a bit surprised that you hadn't already taken this step).

2) Make the data make your analysis more accessible to the community. Perhaps this means incorporating the analysis into one of the website / databases that your lab is involved in. Or perhaps it is a summary table that conveniently summarizes the analysis of the paper. I know that S1 has some of this, but is still not easy for users to query.

3) Is it possible to do something to take it to the next step? Essentially what you have so far is a compilation or review of available data. Can you draw some kind of conclusion from your analysis; is there anything unexpected from the expression data? etc. Another possibility: are the tandem duplications unique to Rice or shared in other grasses? See Gerard's comments for further ideas.

I also have a concern:

4) For the gene expression data you provide a FC and FDR threshold for what is considered differentially expressed, but you do not provide thresholds for the other data sets. Without FDR cutoffs there is no way to know if the differences you describe in the heat maps is meaningful. You applied the same threshold the please stated that. Otherwise correct the analysis. When my lab has been doing some kind of metanalysis of expression as you are doing here we found it best to start with the raw data from each experiment and do our own analysis to determine differentially expressed genes. That way we know that the same pipleine has been used throughout the paper and that analyses are more comparable. It would be a good approach to use here.

And minor point:

5) Line 151. Did you mean "< five genes in between" instead of "> five"?

Gerard's comments:

"This is a confusing manuscript in that the authors go to good depths to demonstrate that there is a wealth of information available to detail and characterize the 144 member SDRLK family by pointing to tissue specific expression patterns with potential interaction attributions. The manuscript appears to show a care in the biocurational approach, yet the reader is still in a place with little to get started. There appear viable descriptors provide in Table S1, yet one would expect addition annotation relating to evolving ontology repertoire, and perhaps include additional information relating toward possible evolutionary structure. There is a good pointers to dataset sources (Table S2); however, the reader still would need to process the information from the beginning to place it into an annotated form with utility. I am confused here that the authors would lead a reader up to a point in a fashion without expanding to provide available tools to actually curate and annotate the data. Why are ontology terms not provided or described, and why are not resources pointed to that would expand for the reader a method to try to apply the finding described. Yes, there is curation, but not in a form the reader to act upon; it seems more of an eloquent way to say they the authors know the data is out there, and provide a road map, but do not actually perform an experiment to push the field forward. An reader would have to start from the very same point to begin this process. Journal manuscripts are often scanned by text-mining software that locates and extracts core data elements, like gene function. Adding standard ontology terms, such as the Gene Ontology (GO, geneontology.org) or others from the OBO foundry (obofoundry.org) can enhance the recognition of your contribution and description. This will also make human curation of literature easier and more accurate. None of this was visible. I would recommend moderate revision to have the context of the manuscript to have more bite,, rather than just highlighting data that already exists. Suggested forms might be to illustrate phylogenetic comparisons, emulate differential expression, or perhaps highlight a unique feature-for-function not already uncovered from data resources used as the roadmap for this presentation."

---

## Round 0.4 · accepted · Accept

Thanks for the revisions. I like the new focus!